# Risk factors for community-acquired respiratory infections in a non-pandemic context: Secondary analysis of the PRIMIT study

Ashley Hammond[1]*, Beth Stuart[2], Paul Little[2], Alastair D. Hay[1]

**1** Centre for Academic Primary Care, University of Bristol, Bristol, England, **2** Primary Care, Population Sciences and Medical Education, University of Southampton, Aldermoor Health Centre, Southampton, England

\* ashley.hammond@bristol.ac.uk

**Data Availability Statement:** The data underlying the results presented in the study are available from J.Cousins@soton.ac.uk; address: Aldermoor

## Abstract

### Objectives

Respiratory tract infection (RTI) incidence varies between people, but little is known about why. The aim of this study is therefore to identify risk factors for acquiring RTIs.

### Methods

We conducted a secondary analysis of 16,908 participants in the PRIMIT study, a pre-pandemic randomised trial showing handwashing reduced incidence of RTIs in the community. Data was analysed using multivariable logistic regression analyses of self-reported RTI acquisition.

### Results

After controlling for handwashing, RTI in the previous year (1 to 2 RTIs: adjusted OR 1.96, 95% CI 1.79 to 2.13, p<0.001; 3 to 5 RTIs: aOR 3.89, 95% CI 3.49 to 4.33, p<0.001; $\geq$6 RTIs: OR 5.52, 95% CI 4.37 to 6.97, p<0.001); skin conditions that prevent handwashing (aOR 1.39, 95% CI 1.24 to 1.55, p<0.001); children under 16 years in the household (aOR 1.27, 95% CI 1.12, 1.43, p<0.001); chronic lung condition (aOR 1.16, 95% CI 1.02 to 1.32, p = 0.026); female sex (aOR 1.10, 95% CI 1.03 to 1.18, p = 0.005), and post-secondary education (aOR 1.09, 95% CI 1.02 to 1.17, p = 0.01) increased the likelihood of RTI. Those over the age of 65 years were less likely to develop an infection (aOR 0.89, 95% CI 0.82 to 0.97, p = 0.009). Household crowding and influenza vaccination do not influence RTI acquisition. A post-hoc exploratory analysis found no evidence these subgroups differentially benefited from handwashing.

### Conclusions

Previous RTIs, chronic lung conditions, skin conditions that prevent handwashing, and the presence of household children predispose to RTI acquisition. Further research is needed

Health Centre, Aldermoor Close, Southampton, SO16 5ST.

**Funding:** PRIMIT was funded by the Efficacy and Mechanism Evaluation Programme (09/800/22), in partnership between the MRC and NIHR. ADH is supported by a NIHR Senior Investigator Award (NIHR 200151), which funded this secondary data analysis. The views expressed in this publication are those of the author(s) and not necessarily those of the MRC, NHS, NIHR, or the Department of Health and Social Care.

**Competing interests:** The authors have declared that no competing interests exist.

to understand how host and microbial factors explain the relationship between previous and future RTIs.

## Introduction

Respiratory tract infections (RTIs) are the most common infection managed by healthcare systems worldwide. RTIs cost the UK economy around £1.7 billion per annum, mainly due to lower RTIs such as pneumonia [1]. RTIs also cause considerable morbidity including loss of earnings due to sickness absence, with 38 million working days lost in 2018 [2].

On average, adults will have two to five RTIs annually, often the 'common cold' or upper respiratory infections [3]. RTIs are the most common indication for antibiotic prescribing [4], accounting for around 60% of all antibiotics prescribed in primary care [5]. This is often driven by patient expectations [6] and rarely provides clinical benefit to the patient given that RTIs are often viral, and/or self-limiting [7, 8].

Despite this, only a small number of studies have investigated risk factors for community-acquired RTIs in non-pandemic lockdown contexts (*BMC Infect Dis* 2021, submitted). Predisposition to acute infection is likely influenced by multiple factors, including social, behavioural, and environmental. The aim of the present study was to use pre-pandemic data to investigate risk factors for acquisition of community-acquired RTIs in England. We used data from the PRIMIT study, a very large randomised controlled trial demonstrating the effectiveness of an internet-delivered handwashing intervention in reducing the incidence of RTIs in the community between January 2011 and March 2013 [9]. PRIMIT provides an ideal data set to assess both the risk factors for acquiring infections, and whether those who are at risk benefit similarly from the intervention.

## Materials and methods

### Study design and setting

Data collection was conducted as previously described [9]. Briefly, the trial enrolled 20,066 patients aged 18 years or over recruited via 344 primary care practices across England. Exclusion criteria included patients with severe mental problems (e.g. major uncontrolled depression, dementia or severe mental impairment), those who were terminally ill and those reporting a skin complaint which would limit their handwashing. Participants were recruited over three autumn, winter and spring periods (January to March 2011; November 2011 to April 2012; and October 2012 to March 2013), with a 16-week follow-up period.

At the point of consent, 18,622 participants were randomly assigned in a 1:1 ratio to receive access to a web-based intervention, which included a baseline questionnaire about current handwashing practices (n = 9350), or no access to the intervention and no baseline questionnaire (n = 9272). In order to understand whether there was a priming effect of asking about handwashing at baseline, an additional cohort of 1444 participants were also randomly assigned in a 1:1 ratio to receive either a control (no intervention but access to the baseline handwashing questionnaire, n = 754), or intervention (access to web-based intervention but no baseline handwashing questionnaire, n = 690). This gave a total of 10,040 participants assigned to the intervention and 10,026 assigned to control.

The study was conducted by researchers at the University of Southampton, in collaboration with Universities of Oxford, Birmingham and Glasgow, trial registration number ISRCTN75058295.

## Outcomes

The primary outcome was binary, defined as participants with one or more reported infections versus no reported infections at 16 weeks. Illnesses were classified as RTIs on the basis of consensus from previous studies [10, 11], defined as two symptoms of an RTI for at least one day or one symptom for two consecutive days. Influenza-like illness was defined as a high temperature (>37.5˚C), a respiratory symptom (sore throat, cough or runny nose), and a systemic symptom (headache, severe fatigue, severe muscle-aches, or severe malaise).

## Data collection

Episodes of infection and their duration were self-reported by study participants. All participants were sent invitations to complete the online outcome assessment measures monthly (at 4, 8, 12, and 16 weeks after initial login). Of the 10,040 study participants who were in the intervention groups, 8241 (82%) completed the follow-up questionnaire at 16 weeks. Of the 10,026 study participants assigned to the control groups, 8667 (86%) completed the 16-week follow-up questionnaire.

## Statistical analysis

For this secondary data analysis, we conducted multivariable logistic regression analysis to explore risk factors for the acquisition of one or more RTIs, as self-reported by study participants over the 16-week follow-up. While this was a complete case analysis of the PRIMIT outcome data, they could only be included in the analysis if they provided follow-up data (at 16 weeks) and did not have any missing data on the covariates we included in our model. Univariable and multivariable odds ratios, and their corresponding 95% confidence intervals, were reported for each covariate, which included age, sex, education after the age of 10, number of people living in the household, children under 16 years living in the household, any ongoing health problems, number of RTIs in previous year, had influenza vaccination in the current season, and skin condition which could affect handwashing. The model also controlled for randomisation group in the trial. P-values were reported for the multivariable logistic regression model only. A post-hoc subgroup analysis was conducted to investigate for differential intervention effectiveness in the newly identified risk groups individually and together, by assigning one point per factor, using the appropriate interaction terms.

## Results

A total of 16,908 study participants were included in our secondary data analysis. Table 1 summarises the number of observations for each of the risk factors of interest. More than 70% of participants were aged 65 years or younger, and 56% were female. Seventy percent of participants reported no ongoing health problems at the time of recruitment.

The multivariable logistic regression analysis provides evidence that female sex, post-secondary education, children younger than 16 living in the household, chronic lung condition, having had at least one RTI in the previous year and having a skin condition that could affect handwashing all increased the likelihood of acquiring an RTI (Table 2). There was some suggestion that those over age 65 were less likely to develop infections (adjusted OR 0.89, 95% CI 0.82 to 0.97, p-value 0.009).

We found no evidence of differential intervention effectiveness in any of the newly identified risk factors, either individually or combined (Table 3). Relative intervention effect sizes were similar when comparing the overall study population with those in the subgroups. When

**Table 1. Number of PRIMIT study participants for each investigated risk factor.**

| Risk factor | Number of study participants with 16-week follow up data (%) | Number of study participants with at least 1 RTI during the study period (%) | Number of study participants without at least 1 RTI during the study period (%) |
|---|---|---|---|
| **Age** | | | |
| *65 and under* | 11976 (71.0%) | 6985 (58.3%) | 4991 (41.7%) |
| *Over 65* | 4895 (29.0%) | 2380 (48.6%) | 2515 (51.4%) |
| **Sex** | | | |
| *Male* | 7422 (44.0%) | 3881 (52.3%) | 3541 (47.7%) |
| *Female* | 9449 (56.0%) | 5484 (58.0%) | 3965 (42.0%) |
| **Education after the age of 10** | | | |
| *Secondary* | 9,486 (57.7%) | 5085 (53.6%) | 4401 (46.4%) |
| *Post-secondary* | 6,949 (42.3%) | 4075 (58.6%) | 2874 (41.4%) |
| **How many people in the household, Mean (SD)** | 2.5 (0.89) | 2.6 (0.94) | 2.4 (0.84) |
| **Children younger than 16 living in household** | 2676 (16.2%) | 1767 (66.0%) | 909 (34.0%) |
| **No ongoing health problems** | 11,461 (70.0%) | 6431 (56.1%) | 5030 (43.9%) |
| **Number of RTIs in previous year** | | | |
| *0* | 3148 (19.2%) | 1171 (37.2%) | 1977 (62.8%) |
| *-1-2* | 9055 (55.1%) | 4943 (54.6%) | 4112 (45.4%) |
| *3–5* | 3726 (22.7%) | 2651 (71.2%) | 1075 (28.9%) |
| *6+* | 507 (3.1%) | 394 (77.7%) | 113 (22.3%) |
| **Had flu vaccination in the current season** | 6593 (40.0%) | 3571 (54.2%) | 3022 (45.8%) |
| **Skin condition prior to the study that could affect handwashing** | 1825 (13.3%) | 1228 (67.3%) | 597 (32.7%) |

one point was assigned for each of risk factors, the interaction term for the resulting score was 0.97 (0.94, 1.01; p = 0.134).

## Discussion

### Summary of main findings

In the largest and most rigorously pre-pandemic study of its kind to date, we found the number of RTIs in previous year, skin conditions that prevent handwashing, living with children under 16 years, female sex and post-secondary education each independently predisposed individuals to acquire a RTI. Neither household crowding nor flu vaccination were independently associated with an increased risk of RTI acquisition.

### Strengths and limitations

To our knowledge, this is the largest study to investigate for factors predisposing to RTI acquisition, thereby reducing type II error. The PRIMIT study was a prospectively designed randomised trial with data collection supported by monthly symptom diary follow-up, which minimises any recall bias associated with self-reported outcomes. However, in order to minimise participant burden, there were several unmeasured factors which may be important in predisposition to acquiring RTIs, including occupation, exercise, body mass index, diet, alcohol intake, smoking, use of herbal or over-the-counter treatments, and recent use of antibiotics.

Furthermore, since participants self-reported RTIs, it is possible that recollection of previous RTIs sensitised recognition of subsequent RTIs. The self-reported nature of the study

**Table 2. Crude and adjusted odds ratios for risk factors associated with acquisition of a respiratory infection during 16-week PRIMIT study period.**

| Predisposing factors | Univariable odds ratio (95% confidence interval) | Multivariable[a] odds ratio (95% confidence interval) | p-value |
|---|---|---|---|
| **Age** | | | |
| *65 and under* | REF | REF | 0.009 |
| *Over 65* | 0.68 (0.63 to 0.72) | 0.89 (0.82 to 0.97) | |
| **Sex** | | | |
| *Male* | REF | REF | 0.005 |
| *Female* | 1.26 (1.19 to 1.34) | 1.10 (1.03 to 1.18) | |
| **Education after the age of 10** | | | |
| *Secondary* | REF | REF | 0.01 |
| *Post-secondary* | 1.23 (1.15, 1.31) | 1.09 (1.02, 1.17) | |
| **Number of people living in the household[b]** | | | |
| | 1.20 (1.15 to 1.24) | 1.02 (0.97 to 1.07) | 0.490 |
| **Children younger than 16 living in household** | | | |
| | 1.68 (1.54 to 1.83) | 1.27 (1.12, 1.43) | <0.001 |
| **Chronic health problems** | | | |
| *All* | 0.93 (0.86 to 0.99) | 0.96 (0.88 to 1.05) | 0.359 |
| *Chronic lung disease* | 1.26 (1.15 to 1.40) | 1.16 (1.02 to 1.32) | 0.026 |
| **Number of RTIs in previous year** | | | |
| *0* | REFERENCE | REFERENCE | |
| *1–2* | 2.03 (1.87 to 2.21) | 1.96 (1.79 to 2.13) | <0.001 |
| *3–5* | 4.16 (3.76 to 4.61) | 3.89 (3.49 to 4.33) | <0.001 |
| *6+* | 5.89 (4.72 to 7.34) | 5.52 (4.37 to 6.97) | <0.001 |
| **Had flu vaccination in the current season** | 0.90 (0.85 to 0.96) | 0.93 (0.86 to 1.01) | 0.09 |
| **Skin condition that could affect handwashing** | 0.91 (0.87 to 0.94) | 1.39 (1.24 to 1.55) | <0.001 |

[a] In the multivariable model, all estimates are adjusted for the other variables in the model and for randomisation group

[b] Increase in odds for each person additional to the two person minimum, which was an eligibility criterion for the PRIMIT study

means that we do not have any information about the RTI which the participant experienced, including the severity of the infection, the cause of the infection (bacterial or viral), the location of the infection (upper or lower RTI) or the duration. Also, some of the questions that participants were asked, were open to definition by them, for example, skin conditions which impact handwashing. Participants were not provided with a clear definition of what constitutes a skin condition, but were asked to decide on this themselves. Therefore, although our secondary

**Table 3. Subgroup effects for PRIMIT primary outcome.**

| | Interaction term (95% CI) | p-value | Odds ratio for subgroup (95% CI) | p-value |
|---|---|---|---|---|
| **Whole study** | | | 0.71 (0.66, 0.76) | <0.001 |
| **Female** | 0.90 (0.80, 1.03) | 0.120 | 0.67 (0.62, 0.74) | <0.001 |
| **Age 65+** | 0.90 (0.80, 1.01) | 0.056 | 0.71 (0.63, 0.80) | <0.001 |
| **Received flu vaccine in current season** | 0.99 (0.87, 1.14) | 0.941 | 0.71 (0.64, 0.78) | <0.001 |
| **Post secondary education** | 0.96 (0.84, 1.10) | 0.543 | 0.69 (0.62, 0.76) | <0.001 |
| **Skin complaint that impacted handwashing** | 0.93 (0.86, 1.02) | 0.133 | 0.60 (0.49, 0.74) | <0.001 |
| **Children under 16 in the household** | 0.96 (0.80, 1.16) | 0.684 | 0.69 (0.58, 0.82) | <0.001 |
| **Household size (greater than median)** | 0.96 (0.83, 1.11) | 0.582 | 0.69 (0.61, 0.78) | <0.001 |
| **Ongoing health problems** | 1.02 (0.88, 1.18) | 0.788 | 0.72 (0.64, 0.82) | <0.001 |
| **Two or more consultations for RTI in the previous year** | 0.99 (0.85, 1.16) | 0.932 | 0.73 (0.63, 0.84) | <0.001 |

analysis found an association between skin conditions that affect handwashing and increased odds of acquiring a RTI, we do not have any information on the specific skin condition they have, or the extent to which this impacts their handwashing.

### How this fits with existing literature

Our result, that age over 65 may protect against acquiring an RTI, contrasts with the literature suggesting older age increases the risk due to waning immunity and immune defence [12]. However, aging is also associated with increased social isolation, which our analyses suggest could have more impact. In addition, our finding that post-secondary education increases the likelihood of acquiring a RTI also contrasts with much of the literature [13]. It may be that this measurement serves as a proxy for different types or quantities of social interactions, or a proxy for the different threshold at which people self-report RTIs, as this study was reliant on self-reporting of RTI symptoms.

Female sex was associated with an increased likelihood of acquiring a RTI. A systematic review exploring differences between sexes and incidence of RTIs found that females appeared at increased risk of upper RTIs, though males appeared at greatest risk due to higher rates of lower respiratory infections [14]. There are a number of important social and behavioural characteristics which vary between sexes which could influence our findings. Females are more often carers of young children or vulnerable people, which could in itself increase risk of RTI acquisition. Living with young children was also an observed risk factor for RTI acquisition. The ongoing global COVID-19 pandemic has highlighted many of these inequalities between males and females, including increased employment of females in healthcare and social care roles, increased childcare responsibilities at home, and increased difficulty to maintain social distancing [15]. The role of female sex and increased risk of acquiring RTIs warrants further investigation, taking into account social, behavioural and lifestyle characteristics which may influence the association.

In addition to female sex, children under 16 years living in the household was associated with an increased likelihood of acquiring a RTI. Our recently published systematic review identified contact with children as a predisposing factor for acquiring community-acquired pneumonia, though only three studies measured this as a risk factor [13]. There is therefore limited evidence on the significance of contact with children as a risk factor for acquiring a RTI, and what kind and frequency of contact may be important. A 2014 study reported that day-care attendance in the first year of life was associated with more upper RTIs and acute otitis media in children compared to those not attending day-care [16]. This suggests a possible link between increased number of social contacts and increased risk of transmitting and acquiring RTIs. This link has also been demonstrated throughout the COVID-19 pandemic, where the population were asked to socially distance and limit contact with other households in order to introduce transmission of SARS-CoV-2 infection. PRIMIT was conducted pre-COVID-19 when social mixing was the norm, and since social distancing measures have been implemented (alongside other public health interventions including handwashing and wearing face masks) there have been significant reductions in the number of reported RTIs in the community [17].

### Implications for practice, policy and future research

It has become evident from the recent global COVID-19 pandemic and UK national lockdown that there are a number of public health interventions which are effective in reducing transmission of RTIs in the community, including handwashing, social distancing and closure of offices, schools and nurseries. The pandemic had a considerable impact on handwashing,

compared to when the original PRIMIT study was conducted, so we must take this into account when interpreting our findings. However, our recently published systematic review indicates that we still know relatively little about risk factors for community-acquired RTIs, reaffirming the importance of the study findings [13]. Handwashing and general hygiene measures have been identified previously as an effective measure to reduce transmission of respiratory infections [18], however the effectiveness of social distancing has not been properly evaluated pre-COVID-19. It may be that some of these interventions may be effective for use in the future during 'high risk' periods to prevent infection transmission.

We still have a limited understanding of the relative contribution of factors which are associated with increased risk of acquiring and transmitting RTIs, and why some people experience more RTIs than others. Future studies must include the social, behavioural and lifestyle factors which play a role in infection acquisition, including contact with children (work and/ or household), place of work, and exercise–some of which may be modifiable [13]. Better characterisation and measures of the contribution of possible biological and immune factors is also needed, as these are likely to play a role in host susceptibility to infection, and could explain age and sex related differences in susceptibility to infections. Interventions, including those implemented during COVID-19 national lockdowns, must continue to be evaluated for their effectiveness at reducing infection acquisition and transmission.

Finally, research is needed to understand how host and microbial factors explain the relationship between previous and future RTIs, in particular whether it can be explained by biological mechanisms such as altered host immunity or effects on the host microbiome, or psychological mechanisms, such as recall bias, sensitising people to report infections due to previous family or personal experience.

## Conclusions

In the largest and most rigorously conducted study of its kind to date, we found the number of RTIs in previous year, skin conditions that prevent handwashing, living with children under 16 years, female sex, and post-secondary education each independently predispose individuals to acquire a RTI. Further research is needed to understand how host and microbial factors explain the relationship between previous and future RTIs.

## Author Contributions

**Conceptualization:** Ashley Hammond, Beth Stuart, Paul Little, Alastair D. Hay.

**Data curation:** Beth Stuart, Paul Little.

**Formal analysis:** Beth Stuart.

**Funding acquisition:** Beth Stuart, Paul Little.

**Investigation:** Beth Stuart.

**Methodology:** Ashley Hammond.

**Supervision:** Alastair D. Hay.

**Writing – original draft:** Ashley Hammond, Beth Stuart.

**Writing – review & editing:** Paul Little, Alastair D. Hay.

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
