## [Decision Letter · Decision Letter 0]

22 Jun 2022

PONE-D-21-30671Risk factors for community-acquired respiratory infections in a non-pandemic context: secondary analysis of the PRIMIT studyPLOS ONE

Dear Dr. Hammond,

Thank you for submitting your manuscript to PLOS ONE. After careful consideration, we feel that it has merit but does not fully meet PLOS ONE’s publication criteria as it currently stands. Therefore, we invite you to submit a revised version of the manuscript that addresses the points raised during the review process.

The manuscript has been evaluated by three reviewers, and their comments are available below.

The reviewers have raised a number of major concerns. They request improvements to the reporting of methodological aspects of the study, for example to clarify the clinical criteria for the RTI and  the definition of  skin conditions. The reviewers also note concerns about the statistical analyses presented and request re-analyses be completed.

Could you please carefully revise the manuscript to address all comments raised?

We look forward to receiving your revised manuscript.

Kind regards,

Lorena Verduci

Staff Editor

PLOS ONE

Journal Requirements:

2. Please, add the registration number of the original study in the Method section of your manuscript.

“PRIMIT was funded by the Efficacy and Mechanism Evaluation Programme (09/800/22), in partnership between the MRC and NIHR. ADH is supported by a NIHR Senior Investigator Award (NIHR 200151). The views expressed in this publication are those of the author(s) and not necessarily those of the MRC, NHS, NIHR, or the Department of Health and Social Care.”

“PRIMIT was funded by the Efficacy and Mechanism Evaluation Programme (09/800/22), in partnership between the MRC and NIHR. ADH is supported by a NIHR Senior Investigator Award (NIHR 200151). The views expressed in this publication are those of the author(s) and not necessarily those of the MRC, NHS, NIHR, or the Department of Health and Social Care.”

Reviewers' comments:

Reviewer's Responses to Questions

**Comments to the Author**

1. Is the manuscript technically sound, and do the data support the conclusions?

Reviewer #1: Partly

Reviewer #2: Yes

Reviewer #3: Partly

2. Has the statistical analysis been performed appropriately and rigorously? 

Reviewer #1: No

Reviewer #2: Yes

Reviewer #3: Yes

3. Have the authors made all data underlying the findings in their manuscript fully available?

Reviewer #1: No

Reviewer #2: No

Reviewer #3: No

4. Is the manuscript presented in an intelligible fashion and written in standard English?

Reviewer #1: Yes

Reviewer #2: Yes

Reviewer #3: Yes

5. Review Comments to the Author

Reviewer #1: The objective of this manuscript is a straightforward risk assessement via running a multiple/multivariable logistic regression to binary RTI responses from a dataset generated in England. It is actually a secondary data analyses from a previous randomized study called PRIMIT, which was IRB approved. My comments are as follows:

(1) Abstract can be rewritten using the following subsections: (a) Objectives, (b) Methods, (c) Results and (d) Conclusions

(2) Results were generated from a study completed in 2013; not sure what knowledge gain we may have in 2022!

(3) Outcomes: Primary outcome definition is wrong! Maybe the authors meant that the binary outcome means 1 if participants reported >= 1 RTI at 16 weeks; 0 otherwise?

(4) Dataset has a decent number of intervention & control participants, and collected from 344 primary care centers. So, why not the center effect was adjusted for via a GEE-type approach, or a random intercept logistic regression model?

(5) Logistic regression Goodness-of-fit, say via Hosmer-Lemeshow statistics, were not presented.

Reviewer #2: I see that this simple yet rigorous manuscript has adequately explained the whole idea of restriction of social engagements could be protective in a sense of risk of contracting RTI. The clinical criteria for its subject may seem to be too permissive, though; e.g. there is no explanation of the severity of the RTI (new, recurrent, acute, an acute episode of a chronic, or debilitating RTI), there is no description of the location of RTI (upper, lower, or mixed), and there is no description of the duration of the RTI, as there are many kinds of RTI which has different outcome. This should be addressed in the manuscript, at least in a brief text.

Reviewer #3: This manuscript is a secondary analysis of the PRIMIT Study and shows risk factors associated with the incidence of RTIs.

This is well written and illustrated paper which presents the results from a large self-reported survey, and the relationship between acquisition of RTIs and risk factors described is more straightforward.

I have some suggestions for improvement;

1. Regarding the results of "education after the age of 10" (aOR 1.09, 95% CI 1.02 to 1.17, p=0.01) shown in Table 2, the text states that years of education do not affect the acquisition of RTI, but it is necessary to discuss this issue further.

2. Regarding the point that older adults are less likely to acquire RTIs, as noted in the text, there is a discrepancy with the findings in the previous literature. The authors should discuss this point.

3. The definition of “skin condition that impacted handwashing” should be described in the text, because it is one of the important findings in this paper.

6. PLOS authors have the option to publish the peer review history of their article (what does this mean?). If published, this will include your full peer review and any attached files.

Reviewer #1: No

Reviewer #2: **Yes: **Irandi Putra Pratomo

Reviewer #3: No

---

## [Author Response · Author response to Decision Letter 0]

26 Aug 2022

Please see our response to reviewers document attached which provides a detailed response to each reviewer comment and highlights where the text in the manuscript has been changed if necessary.

---

## [Decision Letter · Decision Letter 1]

24 Oct 2022

Risk factors for community-acquired respiratory infections in a non-pandemic context: secondary analysis of the PRIMIT study

PONE-D-21-30671R1

Dear Dr. Hammond,

We’re pleased to inform you that your manuscript has been judged scientifically suitable for publication and will be formally accepted for publication once it meets all outstanding technical requirements.

Kind regards,

Joseph Donlan

Staff Editor

PLOS ONE

Additional Editor Comments (optional):

Reviewers' comments:

Reviewer's Responses to Questions

**Comments to the Author**

1. If the authors have adequately addressed your comments raised in a previous round of review and you feel that this manuscript is now acceptable for publication, you may indicate that here to bypass the “Comments to the Author” section, enter your conflict of interest statement in the “Confidential to Editor” section, and submit your "Accept" recommendation.

Reviewer #1: All comments have been addressed

Reviewer #3: All comments have been addressed

2. Is the manuscript technically sound, and do the data support the conclusions?

Reviewer #1: Yes

Reviewer #3: Yes

3. Has the statistical analysis been performed appropriately and rigorously? 

Reviewer #1: Yes

Reviewer #3: Yes

4. Have the authors made all data underlying the findings in their manuscript fully available?

Reviewer #1: Yes

Reviewer #3: No

5. Is the manuscript presented in an intelligible fashion and written in standard English?

Reviewer #1: Yes

Reviewer #3: Yes

6. Review Comments to the Author

Reviewer #1: (No Response)

Reviewer #3: (No Response)

7. PLOS authors have the option to publish the peer review history of their article (what does this mean?). If published, this will include your full peer review and any attached files.

Reviewer #1: No

Reviewer #3: **Yes: **Yasunori Ichimura

---

## [Editor Report · Acceptance letter]

9 Nov 2022

PONE-D-21-30671R1 

Risk factors for community-acquired respiratory infections in a non-pandemic context: secondary analysis of the PRIMIT study 

Dear Dr. Hammond:

I'm pleased to inform you that your manuscript has been deemed suitable for publication in PLOS ONE. Congratulations! Your manuscript is now with our production department. 

Kind regards, 

on behalf of

Dr Joseph Donlan 

Staff Editor

PLOS ONE